# Reaching the ultimate energy resolution of a quantum detector

Bayan Karimi[1]*, Fredrik Brange[2], Peter Samuelsson[2] & Jukka P. Pekola[1]*

Quantum calorimetry, the thermal measurement of quanta, is a method of choice for ultra-sensitive radiation detection ranging from microwaves to gamma rays. The fundamental temperature fluctuations of the calorimeter, dictated by the coupling of it to the heat bath, set the ultimate lower bound of its energy resolution. Here we reach this limit of fundamental equilibrium fluctuations of temperature in a nanoscale electron calorimeter, exchanging energy with the phonon bath at very low temperatures. The approach allows noninvasive measurement of energy transport in superconducting quantum circuits in the microwave regime with high efficiency, opening the way, for instance, to observe quantum jumps, detecting their energy to tackle central questions in quantum thermodynamics.

---

[1] QTF Centre of Excellence, Department of Applied Physics, Aalto University School of Science, P.O. Box 13500, FI-00076 Aalto, Finland. [2] Department of Physics and NanoLund, Lund University, Box 188, SE-221 00 Lund, Sweden. *email: bayan.karimi@aalto.fi; jukka.pekola@aalto.fi

Almost a century ago, Johnson and Nyquist[1,2] presented evidence of fluctuating electrical current and the governing fluctuation dissipation theorem (FDT). Whether, likewise, *temperature T* can fluctuate is a controversial topic and has led to scientific debates for several decades[3–7]. Consider a system with coupling to a heat bath at temperature $T$ for which the classical FDT of fluctuations $S_Q^{eq}$ of heat current $\dot{Q}$ holds in form $S_Q^{eq} = 2k_B T^2 G_{th}$ in equilibrium. Here, $G_{th}$ is the heat conductance to the bath. We can write the energy balance equation $\dot{Q} = \mathcal{C} d\tilde{T}/dt$ for the temperature of the system $\tilde{T}(t) = T + \delta T(t)$ at time $t$, where $\mathcal{C}$ denotes the heat capacity. The heat current is composed of its expectation value $-G_{th}\delta T$ and fluctuations $\delta \dot{Q}$ around it. There are two origins of noise in this heat current: (1) the standard randomness of transport known for particle current noise (time randomness), and (2) random energies exchanged, leading to enhancement of fluctuations on top of those known for particle current only. We obtain the noise spectrum of temperature of the system by Fourier transformation as $S_T(\omega) = \int dt e^{i\omega t} \langle \delta T(t) \delta T(0) \rangle$. This yields under steady state conditions

$$S_T(\omega) = \frac{2k_B T^2}{G_{th}} \frac{1}{1 + \omega^2 \mathcal{C}^2/G_{th}^2}. \quad (1)$$

At low frequencies we have

$$S_T(0) = 2k_B T^2/G_{th}, \quad (2)$$

and the spectrum has Lorentzian cutoff at $\omega_c = G_{th}/\mathcal{C}$. These results hold also for a system coupled to several equilibrium baths, if one takes $G_{th}$ to represent the sum of all the individual thermal conductances to these baths. For the root-mean-square (rms) fluctuations we obtain the well-known result[3] $\langle \delta T^2 \rangle = \int_{-\infty}^{\infty} \frac{d\omega}{2\pi} S_T(\omega) = k_B T^2/\mathcal{C}$.

Here, we measure the time-dependent temperature of the absorber of a nano-calorimeter at low mK temperatures both under equilibrium and nonequilibrium conditions. We observe that the equilibrium fluctuations follow the fluctuation dissipation theorem (FDT) for temperature. Ideally, the noise of this calorimeter permits measurements of microwave photons in GHz regime at the lowest temperatures that we achieve. This method is then a way to observe calorimetrically, e.g., the quantum trajectories with superconducting circuits[8–10].

## Results

**The calorimeter.** In a fermionic system, like the electrons (about $10^8$ of them) in the nano-calorimeter in the present experiment, temperature is coded in the Fermi distribution $f(\epsilon) = [e^{(\epsilon-\mu)/k_B T} + 1]^{-1}$, which directly determines the readout signal of our thermometer. Here, $\epsilon$ and $\mu$ denote the single particle energy and chemical potential of the system, respectively. We illustrate the calorimeter[11–15] principle of our experiment and set-up in Fig. 1[16]. The electron system (absorber), is coupled to the phonon heat bath at constant temperature $T$ via electron–phonon collisions, which lead to stochastic exchange of heat, as indicated by the many vertical arrows between the two in Fig. 1a. This forms the bottleneck of heat transport in a nano-calorimeter, in contrast to macroscopic calorimeters. The red arrows from the left depict the electronic injection of heat under nonequilibrium conditions, fluctuating due to the stochastic nature of tunneling. By attaching a fast thermometer to the absorber, one records its time $t$ dependent temperature fluctuations $\delta T(t)$ as shown by a measured time trace. The actual sample (scanning electron micrograph in Fig. 1b) is realized as a $\ell = 1$ μm long copper normal-metal absorber (brown) connected to three superconducting leads (blue). The right one is a tunnel contact of the thermometer and the other tunnel junction on the left the hot electron injector. The third one pointing down and 50 nm away from the thermometer, is a direct clean metal-to-metal contact grounded at the sample stage. It provides a fixed chemical potential for the absorber and induces proximity superconductivity to the thermometer facilitating its proper operation. The measuring set-up for the thermometer junction shown on the right side of Fig. 1b consists of a parallel on-chip $LC$ resonator, coupled to input $V_1$ and output $V_2$ RF (radio frequency) lines, operating at frequency $f_0 = 620$ MHz, which also admits DC biasing at voltage $V_{th}$. The measured signal $S_{21}$ obtained from the ratio of $V_2/V_1$ yields the conductance of the thermometer junction. It is measured at a finite sampling rate in order to acquire statistics of temporal temperature of the absorber.

**Principles of the experiment.** In order to calibrate the thermometer we measure $S_{21}$ averaged over typically 1 s time interval at different bath temperatures of the cryostat, traceable to primary

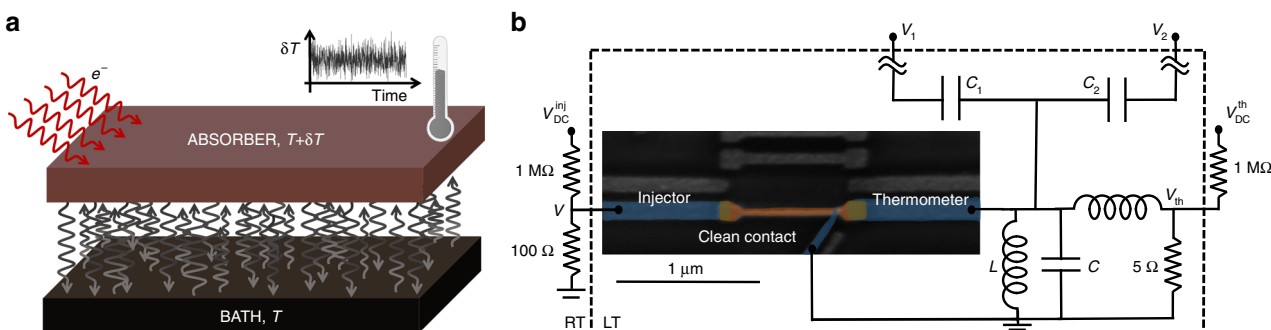

**Fig. 1 The set-up for measuring temperature fluctuations. a** The calorimeter principle applied to the electronic system in this work. The normal-metal absorber in the center is subjected to the fluctuating heat current from the phonon bath below. Additionally we have an option to create nonequilibrium by injecting "hot" electrons as indicated by red arrows on the left. A key element in the calorimeter is a thermometer with sufficient bandwidth to provide temporal temperature traces, of which an example is shown above the absorber. **b** The measurement set-up including the colored scanning electron micrograph of the sample in the center. The $\ell = 1$ μm long Cu absorber (brown) coupled to two superconducting Al leads (blue) via tunnel barriers (bronze). The clean metal-to-metal contact to another superconducting Al lead pointing down at an inclined angle provides the proximity effect for the thermometer and a fixed chemical potential for the absorber. The circuit on the sample stage at low temperature (LT) within the dashed area presents the RF readout of the thermometer junction with tunnel resistance 18 kΩ composed of an LC resonator and probed by RF transmission measurement between ports $V_1$ and $V_2$. The rest of the set-up at room temperature (RT) is for DC biasing of both the injector ($V$) and thermometer ($V_{th}$).

Coulomb blockade thermometry CBT. An example of dependence of thus obtained averaged $\langle S_{21}\rangle$ on $V_{th}$ is shown on a wide bias range in Fig. 2a. The drop of $\langle S_{21}\rangle$ at about $\pm 200$ μV is due to the superconducting gap $\Delta$ in aluminum. The main feature, the zero bias anomaly (ZBA) at $V_{th} = 0$, which is indicated by the central red arrow, presents the basis of our thermometer. This dip originates from proximity induced supercurrent due to the presence of clean contact. Now it is placed 50 nm away from the tunnel junction, which is to be contrasted to 500 nm in our earlier work[17]; this way the sensitivity of the thermometer is enhanced substantially. Quantitatively, the temperature dependence of the average transmission $\langle S_{21}\rangle$ at this dip is depicted in Fig. 2b. It manifests approximately linear dependence at sub 200 mK temperatures, emphasized by the zoom in the inset of this figure. Owing to the competing quasiparticle tunneling, there is eventually back-bending of the characteristics at temperatures above 300 mK; this leads to loss of sensitivity in the cross-over temperature range. Depending on the range of interest, we employ either linear or nonlinear calibration to convert $\langle S_{21}\rangle$ to temperature. This calibration needs to be done only once for each cooldown.

**Equilibrium fluctuations**. Time domain measurements allow detecting temporal fluctuations of the quantity of interest. In our case we monitor $S_{21}(t)$, yielding the instantaneous temperature of the absorber at 10 kHz sampling rate over a chosen time interval. We collect data under given conditions typically for up to 1 hour. As a result we obtain the total fluctuations (variance) $\langle \delta S_{21,tot}^2\rangle$ in a bandwidth of $\Delta f \approx 10$ kHz. This signal is composed of the amplifier and other instrumental noise $\langle \delta S_{21,bg}^2\rangle$ ("bg" stands for background), in addition to the noise of interest from the actual sample, $\langle \delta S_{21}^2\rangle = \langle \delta S_{21,tot}^2\rangle - \langle \delta S_{21,bg}^2\rangle$. Here, we assume uncorrelated noise from the different sources. The way we determine the $\langle \delta S_{21,bg}^2\rangle$ is explained in the Methods section. Our quantitative results depend critically on the precision of determining this background noise. Taking the linear calibration as in the inset of Fig. 2b, with the responsivity $\mathcal{R} \equiv |d\langle S_{21}\rangle/dT|$, we have for the temperature noise of the absorber $\langle \delta T^2\rangle = \mathcal{R}^{-2}\langle \delta S_{21}^2\rangle$. We exhibit in Fig. 3 the central quantity in the experiment, low-frequency temperature fluctuations $\sqrt{S_T} = \sqrt{\langle \delta T^2\rangle/2\Delta f}$ as a function of bath temperature in equilibrium. From now on we denote NET $\equiv \sqrt{S_T}$, which is the noise-equivalent temperature. The data symbols in both panels

correspond to the averaged bare noise, where the best guess of the background has been subtracted. The shaded area in Fig. 3a depicts the uncertainty in determining NET precisely due to this subtraction. Overall, we observe first increase of NET upon lowering $T$ and then gradual turn down of it at the lowest temperatures. The dominant contributions to $G_{th}$ arise from

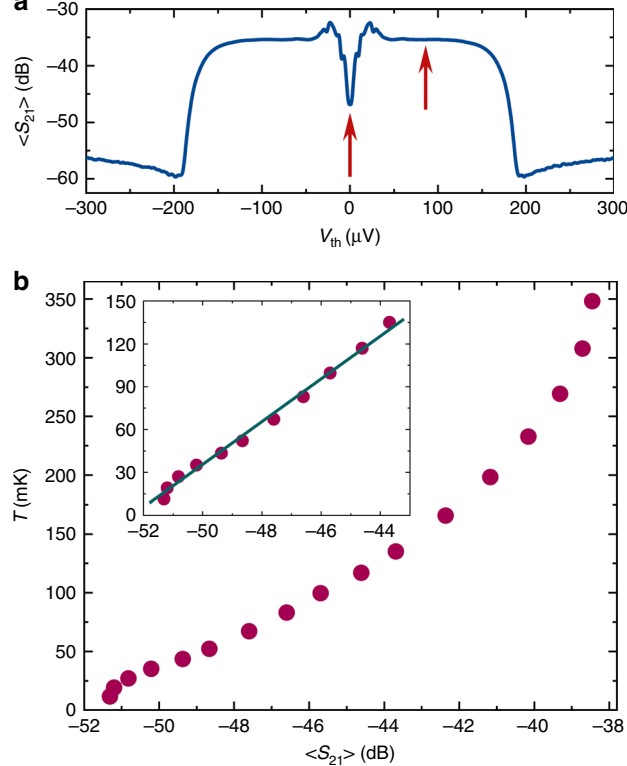

**Fig. 2 The transmission measurement of the RF thermometer at −120 dBm input power. a** Wide bias range transmission $\langle S_{21}\rangle$ averaged over 100 repetitions at each bias point $V_{th}$ at bath temperature $T \sim 100$ mK. The two red arrows indicate the working points for actual ZBA thermometry at $V_{th} = 0$ and background measurement at $V_{th} = 85$ μV, respectively. **b** The thermometer calibration against the bath temperature $T$ in equilibrium. The inset shows the low-temperature end together with the linear fit used for the temperature fluctuation measurements.

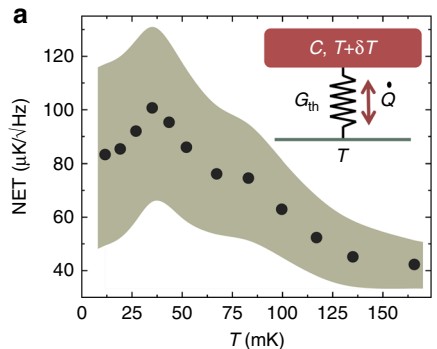
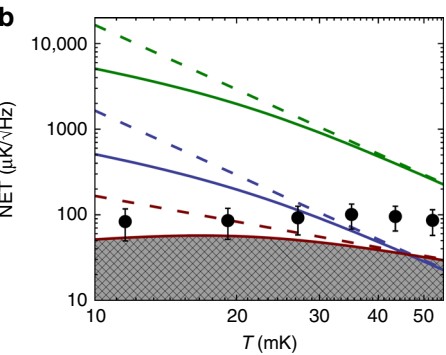

**Fig. 3 Temperature fluctuations in equilibrium. a** Measured low-frequency fluctuations NET $= \sqrt{S_T}$ at different bath temperatures. The symbols are the measured data (both in **a** and **b**) with the mean instrumental background noise subtracted as described in the Methods section. The shaded area covers the uncertainty in this subtraction process. The inset shows schematically the thermal model including the absorber with finite heat capacity $\mathcal{C}$ coupled to the heat bath at temperature $T$ via thermal conductance $G_{th}$. **b** Comparison of the data to the predicted noise-equivalent temperature NET, now on a logarithmic scale, in the absence ($\alpha = 0$, dashed lines) and presence ($\alpha = 10^{-4}$, solid lines) of photon contribution assuming a normal-metal absorber. The error bars on the experimental points are based on the same uncertainty as in **a**. Red lines show the fundamental noise-equivalent temperature in equilibrium NET$_{eq} = \sqrt{2k_B T^2/G_{th}}$, and blue and green lines demonstrate NET $= \delta\epsilon/\sqrt{\mathcal{C}G_{th}}$, which is the required NET of the detector to observe a photon with energies $\delta\epsilon = 1\text{K} \times k_B$ and $\delta\epsilon = 10\text{K} \times k_B$, respectively. Then NET/NET$_{eq}$ gives the expected signal to noise ratio of the experiment.

electron–phonon coupling at higher temperatures and radiative heat transfer by thermal photons[18] towards low $T$ as

$$G_{\text{th}} = 5\Sigma\mathcal{V}T^4 + \alpha g T. \tag{3}$$

Here, $\Sigma$, $\mathcal{V}$ are electron–phonon coupling constant[19] and volume of the absorber, respectively. For the photonic contribution[18], $G_Q = gT$ is the quantum of thermal conductance with $g = \pi k_B^2/6\hbar$. We assume the coupling coefficient $\alpha$ to have values $\ll 1$ according to earlier investigations[20]. Equation (2) predicts then

$$\text{NET} = \sqrt{\frac{2k_B}{5\Sigma\mathcal{V}}}T^{-1} \quad (\text{high } T)$$
$$\text{NET} = \sqrt{\frac{2k_B}{\alpha g}}T^{1/2} \quad (\text{low } T), \tag{4}$$

with cross-over between the two regimes with maximum NET at the temperature $T_{\text{co}} = \left(\frac{\alpha g}{10\Sigma\mathcal{V}}\right)^{1/3}$. Using the literature value[21] $\Sigma = 2 \times 10^9$ WK$^{-5}$m$^{-3}$, the measured volume $\mathcal{V} = 1.0 \times 10^{-21}$ m$^3$ and an educated guess $\alpha \sim 10^{-4}$, we obtain a predicted NET versus $T$. Our simple model above predicts a maximum NET $\sim 60$ $\mu$K/$\sqrt{\text{Hz}}$ at $\sim 20$ mK. This NET is within the error bars of the measured signal in Fig. 3a, b at low temperatures. Figure 3b makes a quantitative comparison of the measured sub 50 mK equilibrium noise against the presented model. The solid and dashed red lines indicate NET$_{\text{eq}} = \sqrt{2k_B T^2/G_{\text{th}}}$ with and without the photon contribution using the parameters given above, respectively. The shaded area exhibits the impermissible range due to the fundamental temperature noise in equilibrium. We reach this bound at temperatures well below 30 mK. The rest of the lines in this figure will be discussed later.

The analysis above could be improved, provided the parameters of the system were known precisely. Till now we assumed the absorber to be in the normal state. However, the clean absorber-superconductor contact leads to a proximity induced superconductivity in the absorber. This suppresses the density of states around the Fermi level, on the scale of the Thouless energy $E_{\text{Th}} = \hbar D/\ell^2 \sim 10$ $\mu$eV, resulting in a decreased electron–phonon coupling. Here, $D \sim 0.01$m$^2$/s is the diffusion constant of the Cu film. As a consequence, for electron temperatures below $E_{\text{Th}}/k_B \sim 100$ mK, the thermal conductance $G_{\text{th}}$ is decreased[22] and, hence, the temperature noise NET is increased. The experimentally observed NET $\sim 80$ $\mu$K/$\sqrt{\text{Hz}}$ at low $T$ can then be obtained using $D = 0.01$ m$^2$/s and $\alpha = 10^{-3}$. One should also note that the fluctuations $\delta T$ of temperature become non-negligible as compared to $T$ based on the estimate $\delta T/T \simeq \sqrt{k_B/\mathcal{C}} \gtrsim 0.1$ at $T = 10$ mK for our absorber.

**Nonequilibrium fluctuations.** Let us finally consider the nonequilibrium fluctuations[23–26]. In the measurements presented up to now the injector junction with tunnel resistance $R_T = 20$ k$\Omega$ on the left in Fig. 1b has been unbiased in order to ensure equilibrium. By applying a voltage $V$ to it, the system can be driven into nonequilibrium. The well-known influence of such biasing of a superconductor-normal-metal junction is that it serves as a local refrigerator of the normal-metal absorber thanks to the energy gap of the superconductor, i.e., it acts as an evaporative cooler[27]. This effect is manifested in the bias dependence of the average temperature of the absorber, obtained from the values of $\langle S_{21} \rangle$ in Fig. 4a.

Injecting electrons does not only change the average temperature of the absorber but, due to the stochastic nature of tunneling, it leads to noise of heat current as well[28,29]. Quantitatively this

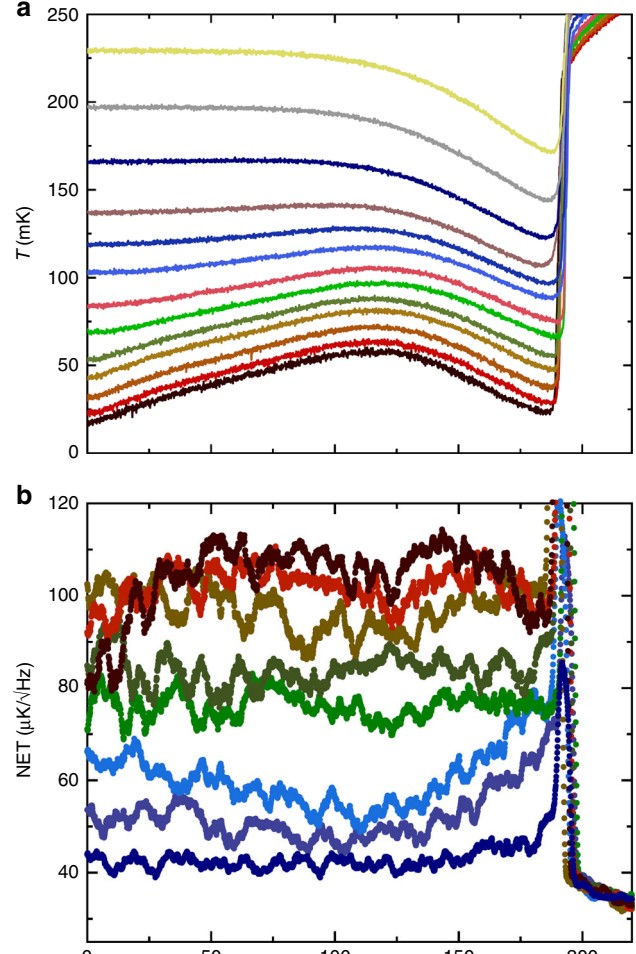

**Fig. 4 Temperature and its fluctuations under nonequilibrium conditions.** **a** Average temperature of the absorber when the injecting junction is biased at different voltages $V$. The data sets correspond to bath temperatures 12, 27, 35, 43, 52, 67, 83, 100, 117, 135, 166, 198, and 233 mK from bottom to top. **b** Nonequilibrium temperature fluctuations at temperatures 12, 27, 35, 52, 67, 100, 117, 166 mK with the same colors as in **a** as a function of injector bias.

noise at low frequencies is given by

$$S_Q^{\text{in}} = \frac{1}{e^2 R_T}\int dE(E - eV)^2 n_S(E)$$
$$\times \{f_N(E - eV)[1 - f_S(E)] + f_S(E)[1 - f_N(E - eV)]\}, \tag{5}$$

where $f_N$, $f_S$ are the energy distribution functions for the normal-metal and superconductor electrons, respectively, and $n_S(E) = |E|/\sqrt{E^2 - \Delta^2}\theta(|E| - \Delta)$ denotes the density of states for the superconductor, with $\theta(x)$ being the Heaviside step function. For typical voltages and temperatures in the regime well below the superconducting gap, the injection noise $\sqrt{S_Q^{\text{in}}}$ is exponentially suppressed[16]. In contrast, the equilibrium noise due to phonons, $\sqrt{S_Q^{\text{eq}}}$, is of a roughly constant magnitude $\sim 10^{-20}$ W/$\sqrt{\text{Hz}}$. Therefore, it is not surprising that the temperature noise in Fig. 4b does not change much at sub-gap voltages $V < 200$ $\mu$V, in particular as the temperature of the absorber is not changing dramatically in this bias range. For these uncorrelated sources the temperature noise

is predicted to obey $S_T = (S_Q^{eq} + S_Q^{in})/G_{th}$. The sudden decrease of temperature noise NET at $V > 200\,\mu V$ is natural since $G_{th}$ increases rapidly when the absorber heats up in this regime (see Fig. 4a). The sharp peak at the gap (Fig. 4b) is possibly an artifact arising from unavoidable voltage noise of the injector, which directly transforms to temperature noise due to the strong voltage dependence of temperature at this point. Yet we find close resemblance of our measured bias-dependent noise and the theoretical predictions by Laakso et al.[26] calculated for a SINIS (superconductor-insulator-normal metal-insulator-superconductor) device.

## Discussion

The temperature that fluctuates is given by the energy distribution of the electrons in the absorber. It qualifies as temperature for the following reasons. (i) Number of particles is large, about $10^8$. (ii) Owing to fast electron–electron internal relaxation over a time scale of $\sim 10^{-9}$ s[30], the carriers form a local Fermi–Dirac distribution: all other relaxation times, most notably the electron–phonon time ($\sim 10^{-5}$ s) are much slower[31]. Furthermore, the temperature of the absorber is spatially uniform, since the heat diffusion time of electrons in the absorber, $\tau_{diff} = \gamma \rho \ell^2 / \mathcal{L}_0 \sim 10^{-10}$ s is very short. Here, $c = \gamma T$ is the specific heat due to conductance electrons with $\gamma \sim 10^2$ Jm$^{-3}$ K$^{-2}$, $\rho \sim 10^{-8}$ $\Omega$m is the resistivity of the Cu, and $\mathcal{L}_0 = 2.44 \times 10^{-8}$ W$\Omega$K$^{-2}$ is the Lorenz number.

A central question is the projected energy resolution of the presented calorimeter. The objective is to use it for observing quanta in the microwave regime. Unlike some of the previously published works on THz calorimetry[32,33], here we aim into the GHz regime common in circuit QED (quantum electrodynamics) experiments. Here, we demonstrated that its resolution is as good as nature can allow, limited only by thermal fluctuations and illustrated by the red lines in Fig. 3b. Indeed, as we present by the blue lines in the figure, the necessary NET of the detector to observe microwave photons, e.g., those emitted by a standard superconducting qubit with $0.5 - 1\,K \times k_B$ energy is well above the fundamental fluctuations at sub 30 mK temperatures.

## Methods

**Background measurements**. We measure the instrumental noise dominated by that of the low-temperature Caltech CITLF2 cryogenic SiGe low-noise amplifier $\langle \delta S_{21,bg}^2 \rangle$ by carefully off-tuning the interesting fluctuations from the sample itself. This is achieved by simultaneously (i) biasing the thermometer junction away from the ZBA regime ($V_{th} \simeq 85\,\mu V$), and (ii) measuring at either below or above the resonance at frequency $f_0$. An example of the corresponding parametric background noise measurement, in form $\sqrt{\langle \delta S_{21,bg}^2 \rangle}$ versus $\langle S_{21} \rangle$ is presented in Fig. 5. We see a typical increase of noise when the attenuation increases towards left. This dependence can be understood quantitatively by assuming constant voltage noise independent of $\langle S_{21} \rangle$. The measured transmission can be written as

$$S_{21} = 20\lg(v/\tilde{v}),\qquad(6)$$

where $v$ is the output of the last stage amplifier, $\tilde{v} = \sqrt{50 \times 1\,mW} \simeq 224$ mV. Noise of $v$ translates then into variations of $S_{21}$ in linear regime as

$$\delta S_{21} = \frac{20}{\ln 10}\frac{\delta v}{v},\qquad(7)$$

and can be written with the help of Eq. (6) for the rms values as

$$\sqrt{\langle \delta S_{21,bg}^2 \rangle} = \frac{20}{\ln 10}\frac{\sqrt{\langle \delta v^2 \rangle}}{\tilde{v}}10^{-\langle S_{21}\rangle/20}.\qquad(8)$$

Based on the fit parameter $a$ in Fig. 5a and the total gain of 60 dB of the amplifier chain, we find the input voltage noise to be $\sim 12$ nV corresponding to the noise temperature of the amplifier of $T_n \sim 5$ K, which is in line with its specifications by the manufacturer.

Figure 5b presents background measurements at frequencies both below and above the resonance over a wide range of attenuation $\langle S_{21} \rangle$. We observe two features that we need to consider when making an accurate evaluation of the $\langle \delta S_{21,bg}^2 \rangle$. First, at large attenuations, due to the fact that the changes are not fully linear in the sense of Eq. (7), the exponential dependence of Eq. (8) is not obeyed strictly. Therefore, we resort to polynomial fits in two regimes, to capture the dependence over the full

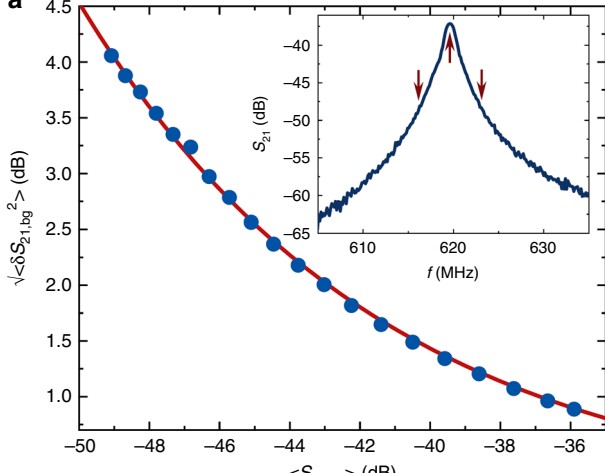

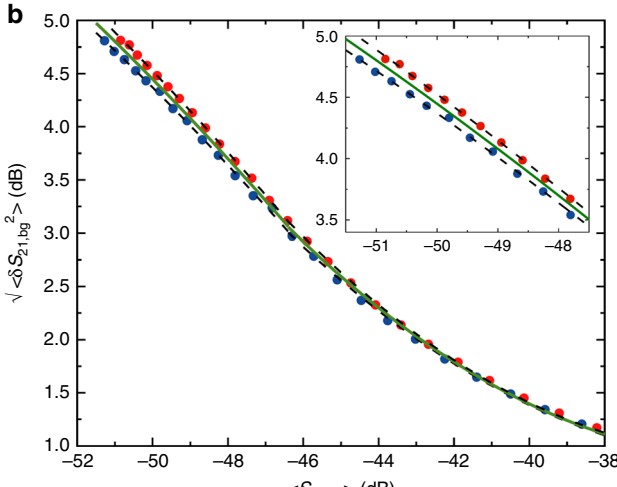

**Fig. 5 Background noise measurements.** All the data are taken outside the zero bias regime of the thermometer and at nonresonant frequencies to exclude the actual noise from the sample. The inset of **a** shows an example of $\langle S_{21} \rangle$ measured around the resonance frequency indicated by the central upward arrow. The blue dots in the main frame of **a** depict parametric plot $\sqrt{\langle \delta S_{21,bg}^2 \rangle}$ versus $\langle S_{21} \rangle$ at the bias voltages $V_{th} = 85\,\mu V$ and at frequencies below the resonance down to 614 MHz indicated by a downward arrow. The red solid line shows the predicted dependence of Eq. (8) yielding the noise temperature of the amplifier of $T_n = 4.9$ K as the only fit parameter of the curve (constant noise voltage at the input). **b** The full range measurement of the background as in **a** but now both above and below the resonance with red and blue dots, respectively. The polynomial fits for the two backgrounds separately (black dashed lines) and the average of them (green solid line) are shown, and they define the mean and the shaded area in Fig. 3. The inset of **b** is simply the zoom-out of the high attenuation range of the main frame.

range. Second, there is a weak dependence of the amplifier noise on frequency; thus the data taken below and above the resonance differ from each other slightly. What we do then, e.g., in Fig. 3, is that we take the mean between the two background measurements as the reference and indicate by the shaded area the uncertainty incurred due to the difference between the two extremes. We thus assume that the frequency dependence of the noise is more or less smooth in the narrow range of $\sim 10$ MHz around $f_0$, and interpolate the data accordingly.

**Experimental details**. The sample (Fig. 1b) was fabricated on standard oxidized Si substrate using Ge process for achieving robust deposition mask[34,35]. The electron-beam lithography was used to pattern the structure for three-angle shadow evaporation of metals. First we deposit 20 nm of Al making the leads followed by oxidation in pure $O_2$ (1 min at 1 mbar). Next another Al layer of 20 nm thickness again provides the clean superconducting contact at the distance of 50 nm from the thermometer

junction, and finally we deposit 35 nm Cu to form the absorber. In the main text we give an estimate of the volume of the absorber based on this thickness; the effective thickness may be somewhat smaller due to the partial oxidation of the film. The resonator is a spiral on a separate chip made of 100 nm thick Al by simple one angle evaporation. The heart of the measuring set-up is shown in Fig. 1b with inductance $L = 100$ nH, $C_1 = 10.3$ fF and $C_2 = 59.3$ fF as coupling capacitors, and $C = 0.2$ pF. The rest of the RF circuitry follows closely to what is presented in ref. [31]. All measurements were performed in a carefully shielded and filtered set-up described in ref. [36].

## Data availability

The data and the numerical code that support the plots within this article are available from the corresponding author upon reasonable request.

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

## Acknowledgements

We acknowledge J. T. Peltonen, E. T. Mannila, O.-P. Saira and S. Gasparinetti for technical support, W. Belzig, D. Nikolic, I. Khaymovich, and T. Tuukkanen for discussions and tests of thermometry, and M. Campisi and K. Saito for useful discussions. This work was funded through Academy of Finland grants 297240, 312057 and 303677 and from the European Union's Horizon 2020 research and innovation program under the European Research Council (ERC) program and Marie Sklodowska-Curie actions (grant agreements 742559 and 766025). F.B and P.S. were supported by the Swedish VR. We acknowledge the facilities and technical support of Otaniemi research infrastructure for Micro and Nanotechnologies (OtaNano).

## Author contributions

The experiment was proposed by J.P. and its realization was conceived by all the authors. B.K. performed the experiment, and designed and fabricated the samples. Data analysis and modeling were performed by B.K. and J.P., with contributions on the noise analysis by F.B. and P.S. The manuscript was written by B.K. and J.P.

## Competing interests

The authors declare no competing interests.
