## [Peer Review File · Nature Communications]

Reviewers' comments:

Reviewer #1 (Remarks to the Author):

Report on manuscript NCOMMS-19-30349-T

The authors present an experiment performed at very low temperature setting the lowest energy (temperature) variation that can be measured on a quantum systems when coupled to a macroscopic heat bath. The temperature fluctuations have been measured and compared to the expected minimum noise in the conditions of the present experimental set-up. It shows that, indeed, the noise floor is reached and this result permits one to envisage the detection of individual microwave photons.

This is an outstanding experiment performed in extreme conditions of temperature and signal to noise ratio. The results are sound and are above state-of-the-art, then the work clearly deserves publication.

Two comments though: 1-What is intrinsically different between a nanosystem and a macroscopic systems regarding the present experiment (except from the small mass, and consequently the small C) ? 2-Regarding nanoscale electron calorimeter, the comparison to anterior work is absent; it would have been interesting to provide a comparison to previous experiments like for instance the one published by Jian WEI et al. Ultrasensitive hot-electron nanobolometers for terahertz astrophysics (Nature Nanotechnology 3, 496 (2008)).

Reviewer #2 (Remarks to the Author):

I could recommend the manuscript entitled "Reaching the ultimate energy resolution of a quantum detector" for publication in Nature Communications for the reason that it presents the first measurement of temperature fluctuations in a SINIS device and that it makes it possible non-invasive observation of energy transport. After a careful calibration of their thermometer, the authors measure the variation of the NET (noise-equivalent temperature) as a function first of temperature and second of voltage. The physical justifications of the global behaviors are given in both cases. The maximum in the NET signal as a function of the temperature is explained by the competing contributions to the thermal conductance (electron-phonon coupling plus thermal photons), and the position of the maximum is evaluated to be 20 mK, in qualitative agreement with their observation. The fact that the NET signal versus the voltage shows fluctuations around a constant value is justified by the suppression of the non-equilibrium noise in the regime below the superconducting gap. However, I can not explain myself why the authors do not cite the paper written by Laasko et al. [Physical Review B 85, 184521 (2012)] in their references list. Indeed, a look to the Figures 3 and 5 of this paper could shed new light on the results presented by the authors in their Figure 4. I consider indeed that a comparison with this paper is the sine qua non condition for this manuscript to be recommended for publication in Nature Communications.

In addition, I have some minor corrections to ask for:

- Add the definition of the function f_S entering in Equation (5).
- Add the definition of the function θ entering in the definition of $n_S(E)$ after Equation (5).
- Line 120: it is not clear why the authors can write that "One should also note that the fluctuations ΔT of temperature become comparable to T itself at the lowest temperature". Is there any figure that shows such a behavior? More explanations are needed on that point.
- Line 150: the expression "relaxation rates" has to be changed to "relaxation times".
- Figure 3b: write somewhere, if not yet done, that a logarithmic scale is used.
- Figure 4 can be improved by :
 - 1) taking the same horizontal scale range for Figure 4a and Figure 4b
 - 2) using the same temperature color codes for Figure 4a and Figure 4b

3) changing θ to NET in the vertical axis legend of Figure 4b in order to be consistent with Figure 3

Reviewer #3 (Remarks to the Author):

Reviewer's report on Manuscript NCOMMS-19-30349-T

TITLE: Reaching the ultimate energy resolution of a quantum detector

AUTHORS: Bayan Karimi et al

The paper is informative, written in a clear and comprehensible manner, its content is within the scope of Nature Communications and, as far as I can tell, original work. Additional clarifications along the points below are suggested that would be, I think, beneficial to readers.

1) The central question of the presented work, as the authors state, is the projected energy resolution of their calorimeter detector aiming at single photons with energies in the range of 1K...10K x kB. However, the detection efficiency would be a relevant device parameter, as well. How do the authors foresee coupling of single photons to the device, and what detection efficiency can be expected, given the presented configuration (materials and dimensions)?

2) The detector operation requires a $\langle S_{21} \rangle$ (T) calibration, which has been performed against a separate thermometer. How stable is this calibration? Is it, for instance, necessary to recalibrate the detector every time it is cooled down to its operation temperature from above T_c of the Al electrodes or from room temperature?

3) In the modelling of their device, the authors assume the Cu absorber element to be entirely metallic. Effects of the proximity-induced superconductivity are discussed. Cu is known to oxidize at surfaces or interfaces in the presence of air or oxide substrates. Thicknesses of native copper oxide layers can be $\sim 3\text{--}6\text{nm}$, hence not negligible given the dimensions on the absorber element. From the Experimental Details it has to be assumed that the absorber element is directly deposited onto a SiO_x layer, and that no top surface passivation layer is present. I suggest to discuss this issue in the context of the device modelling and also with regards to the 'long term' stability of the detector.

4) The description of the measurement setup mentions the devices and the LC resonator chip merely as being placed at 'low temperature'. I assume, that the device is shielded against stray radiation (direct higher temperature thermal radiation or EMI, eg, along the wiring to the device). I suggest to add details on the shielding used or references.

There is a typo in the unit of the gamma value in line 154.

Reviewer #1 (Remarks to the Author):

Report on manuscript NCOMMS-19-30349-T

The authors present an experiment performed at very low temperature setting the lowest energy (temperature) variation that can be measured on a quantum systems when coupled to a macroscopic heat bath. The temperature fluctuations have been measured and compared to the expected minimum noise in the conditions of the present experimental set-up. It shows that, indeed, the noise floor is reached and this result permits one to envisage the detection of individual microwave photons.

This is an outstanding experiment performed in extreme conditions of temperature and signal to noise ratio. The results are sound and are above state-of-the-art, then the work clearly deserves publication.

We thank the referee for the very positive assessment of our work.

Two comments though: 1- What is intrinsically different between a nano-system and a macroscopic systems regarding the present experiment (except from the small mass, and consequently the small C)?

A fundamental difference between a classical macroscopic calorimeter and a nano-calorimeter, as in our case, is as follows. The basic thermal model (C, G_{th},...) is the same in the two cases. However, the thermal coupling in the first case is man-made between different phonon systems. The electrons share the same temperature with phonons in this case naturally. In our nano-calorimeter electrons are the constituent that determine the C and the coupling G_{th} is between the sub-systems, electrons and phonons, arising from a well-defined natural process. Of course the quantitative difference is what the referee pointed out: the mass (heat capacity) is small for a nano-system.

Additionally, we find that at the energy and temperature scales of the experiment, one needs to account for quantum coherent effects via the superconducting proximity effect induced into the normal state absorber, to quantitatively describe the obtained results. The typical spatial extent of the proximity effect in the experiment is on the sub-micron scale, making it negligible in macroscopic systems.

We have added text (highlighted blue) on lines 38 and 45-46 concerning this point.

2-Regarding nanoscale electron calorimeter, the comparison to anterior work is absent; it would have been interesting to provide a comparison to previous experiments like for instance the one published by Jian WEI et al. *Ultrasensitive hot-electron nanobolometers for terahertz astrophysics* (*Nature Nanotechnology* 3, 496 (2008)).

This was indeed largely missing in the manuscript. We have now added the following text in the final paragraph of the main text: "Unlike some of the previously published works on THz calorimetry \cite{martinis, wei}, here we aim into the GHz regime common in circuit QED (quantum electrodynamics) experiments."

The new references are:

M. Nahum and John M. Martinis, Ultrasensitive-hot-electron microbolometer, *Appl. Phys. Lett.* **63**, 3075 (1993).

Jian Wei, David Olaya, Boris S. Karasik, Sergey V. Pereverzev, Andrei V. Sergeev and Michael E. Gershenson, Ultrasensitive hot-electron nanobolometers for terahertz astrophysics, *Nature Nanotechnology* **3**, 496 (2008).

Reviewer #2 (Remarks to the Author):

I could recommend the manuscript entitled "Reaching the ultimate energy resolution of a quantum detector" for publication in Nature Communications for the reason that it presents the first measurement of temperature fluctuations in a SINIS device and that it makes it possible non-invasive observation of energy transport. After a careful calibration of their thermometer, the authors measure the variation of the NET (noise-equivalent temperature) as a function first of temperature and second of voltage. The physical justifications of the global behaviors are given in both cases. The maximum in the NET signal as a function of the temperature is explained by the competing contributions to the thermal conductance (electron-phonon coupling plus thermal photons), and the position of the maximum is evaluated to be 20 mK, in qualitative agreement with their observation. The fact that the NET signal versus the voltage shows fluctuations around a constant value is justified by the suppression of the non-equilibrium noise in the regime below the superconducting gap. However, I can not explain myself why the authors do not cite the paper written by Laakso et al. [Physical Review B 85, 184521 (2012)] in their references list. Indeed, a look to the Figures 3 and 5 of this paper could shed new light on the results presented by the authors in their Figure 4. I consider indeed that a comparison with this paper is the sine qua non condition for this manuscript to be recommended for publication in Nature Communications.

We thank the referee for the positive response and for pointing to the paper of Laakso et al. That paper discusses temperature fluctuations in a SINIS-system, closely related to our SNIS system. Since the contact between the normal state absorber and one of the superconducting terminals in our system is transparent, i.e. not an insulator (I), a quantitative comparison between our results and the predictions by Laakso et al. needs some care. For example, in the SINIS-system electrical potential fluctuations in the absorber are important, while in the SNIS-system they are suppressed, with the potential of the absorber anchored to the chemical potential of the superconductor. Likewise, the nature and relevance of the proximity effect (see response to referee 1) is different in the two systems.

Even taking these differences in the set-up into account, Figures 3 and 5 of the Laakso paper have qualitative similarities with the result in our Figure 4, as the referee points out. This holds in particular for the non-monotonic temperature vs. voltage behavior for low voltage (Fig. 3, Laakso) and the large temperature fluctuations at voltages around the superconducting gap Δ (Fig. 5, Laakso). The strong fluctuations appear at $eV=2\Delta$ in Laakso et al. and not $eV=\Delta$ as in our case, a trivial difference of SINIS vs SNIS.

We included Laakso et al. in our list of references and discuss it in the text. For the discussion of the large temperature fluctuations at voltages around the superconducting gap, we point to the Laakso paper as one possible explanation.

We added "Yet we find close resemblance of our measured bias dependent noise and the theoretical predictions by Laakso et. al. [Laakso] calculated for a SINIS device.", and rephrased the text preceding it.

Ref. Laakso, M. A., Heikkilä, T. T. & Nazarov, Y. V. Theory of temperature fluctuation statistics in superconductor-normal metal tunnel structures, Phys. Rev B **85**, 184521 (2012).

In addition, I have some minor corrections to ask for:

- Add the definition of the function f_S entering in Equation (5).
 - Add the definition of the function θ entering in the definition of $n_S(E)$ after Equation (5).
- Added "f_N, f_S are the energy distribution functions for normal metal and superconductor electrons, respectively, and $n_S(E)=|E|/\sqrt{E^2-\Delta^2}\theta(|E|-\Delta)$ denotes the density of states for the superconductor, with $\theta(x)$ being the Heaviside step function."

- Line 120: it is not clear why the authors can write that "One should also note that the fluctuations δT of temperature become comparable to T itself at the lowest temperature". Is there any figure that shows such a behavior? More explanations are needed on that point.

We thank the referee for pointing out this somewhat sloppy statement. Frankly $\delta T/T$ is still small even at the lowest temperatures, although not quite differential: according to equilibrium fluctuations $\sqrt{\langle \delta T^2 \rangle}/T = \sqrt{k_B/C}$, which at 10 mK yields ≈ 0.1 for a normal absorber, and taking into account the proximity effect it can be larger. We wrote this sentence being somewhat conservative of the estimate. We have changed the sentence into: "One should also note that the fluctuations δT of temperature become non-negligible as compared to T based on the estimate $\delta T/T \simeq \sqrt{k_B/C} \gtrsim 0.1$ at $T=10$ mK for our absorber."

- Line 150: the expression "relaxation rates" has to be changed to "relaxation times". Thank you, it is changed.

- Figure 3b: write somewhere, if not yet done, that a logarithmic scale is used. We added "now on a logarithmic scale" to the caption of Fig. 3b.

- Figure 4 can be improved by :

- 1) taking the same horizontal scale range for Figure 4a and Figure 4b
- 2) using the same temperature color codes for Figure 4a and Figure 4b
- 3) changing Θ to NET in the vertical axis legend of Figure 4b in order to be consistent with Figure 3

We have done all these changes, and indicated in blue the new figure caption.

Reviewer #3 (Remarks to the Author):

Reviewer's report on Manuscript NCOMMS-19-30349-T

TITLE: Reaching the ultimate energy resolution of a quantum detector

AUTHORS: Bayan Karimi et al

The paper is informative, written in a clear and comprehensible manner, its content is within the scope of Nature Communications and, as far as I can tell, original work. Additional clarifications along the points below are suggested that would be, I think, beneficial to readers.

We were very pleased to read the positive judgement of the referee.

- 1) The central question of the presented work, as the authors state, is the projected energy resolution of their calorimeter detector aiming at single photons with energies in the range of 1K...10K x kB. However, the detection efficiency would be a relevant device parameter, as well. How do the authors foresee coupling of single photons to the device, and what detection efficiency can be expected, given the presented configuration (materials and dimensions)?

We refer to the measurement set-up in the attached figure above which is of the type we have used in experiments ([Nature Physics **14**, 991, 2018] and [arXiv:1908.05574, 2019]). The qubit emits a photon to the resonator and it is to be absorbed in the resistor (absorber in the blue frame, our detector used in the present work). The lossy resonator has typically a quality factor of $Q=10$, because of the resistive termination [Appl. Phys. Lett. 115, 022601, 2019]. In the absence of this resistor the quality factor is $Q_0=10^4$ according to the measurement in the same reference. The efficiency to catch the photon is then approximately

$$\eta = \frac{1/Q}{1/Q_0 + 1/Q}$$

which is nearly 100%. Thus we claim that this method has high detection efficiency, but it comes naturally with uncertainty in the actual time instant of the click in the thermometer, due to the stochastic nature of the quantum jump process. The structure presented in the figure above is something that we are currently preparing for experiments for precisely this purpose.

Since all this is still quite speculative until the actual experiment described above will be performed, we simply added “with high efficiency” in the opening paragraph.

2) The detector operation requires a $\langle S_{21} \rangle (T)$ calibration, which has been performed against a separate thermometer. How stable is this calibration? Is it, for instance, necessary to recalibrate the detector every time it is cooled down to its operation temperature from above T_c of the Al electrodes or from room temperature?

As often is the case, the calibration is stable within one cooldown, no recalibration needed, even when one cycles up to the temperature above T_c . But naturally the calibration is sample dependent, and also when thermally cycling up to room temperature, the calibration needs to be repeated, even for the same thermometer.

Now we add “This calibration needs to be done only once for each cooldown.” on page 3.

3) In the modelling of their device, the authors assume the Cu absorber element to be entirely metallic. Effects of the proximity-induced superconductivity are discussed. Cu is known to oxidize at surfaces or interfaces in the presence of air or oxide substrates. Thicknesses of native copper oxide layers can be $\sim 3 \dots 6$ nm, hence not negligible given the dimensions on the absorber element. From the Experimental Details it has to be assumed that the absorber element is directly deposited onto a SiOx layer, and that no top surface passivation layer is present. I suggest to discuss this issue in the context of the device modelling and also with regards to the ‘long term’ stability of the detector.

The referee is right. We intentionally tried to avoid this issue experimentally by immediately cooling down the sample right after the fabrication. Concerning the copper used as an absorber here, the device is not indeed stable, except if keeping it under suitable conditions, meaning vacuum and low temperature. We have added in the section “Experimental details” the following sentence:

“In the main text we give an estimate of the volume of the absorber based on this thickness; the effective thickness may be somewhat smaller due to the partial oxidation of the film.”

4) The description of the measurement setup mentions the devices and the LC resonator chip merely as being placed at ‘low temperature’. I assume, that the device is shielded against stray radiation (direct higher temperature thermal radiation or EMI, eg, along the wiring to the device). I suggest to add details on the shielding used or references.

We do shield and filter the set-up carefully using the methods found effective when studying transport in NIS-type structures. This set-up was optimized in the PhD thesis work in our group cited below. The main issue is to use lossy DC lines and avoid any radiation by hermetically encapsulating the sample stage. This is the key for low electron temperatures and quasiparticle free experiments. Now we added “All measurements were performed in a carefully shielded and filtered set-up described in Ref. [Saira].” in the “Experimental details” section.

Ref. Saira, O.-P. Electrostatic control of quasiparticle transport in superconducting hybrid nanostructures. PhD thesis, Aalto Univ., pp. 41-44 (2013).
[<https://aaltodoc.aalto.fi/bitstream/handle/123456789/8943/isbn9789526050768.pdf?sequence=3&isAllowed=y>]

There is a typo in the unit of the gamma value in line 154.

We thank the referee for pointing out this mistake. Now we corrected it in the text highlighted in blue.

REVIEWERS' COMMENTS:

Reviewer #1 (Remarks to the Author):

The authors have adequately answered the referee's questions and/or comments. for me it is ok for publications.

Reviewer #2 (Remarks to the Author):

In their response to my report, the authors has answered convincingly to all my comments, so that I now recommend the manuscript entitled "Reaching the ultimate energy resolution of a quantum detector" for publication in Nature Communications.

Reviewer #3 (Remarks to the Author):

Reviewer's report on Manuscript NCOMMS-19-30349A

TITLE: Reaching the ultimate energy resolution of a quantum detector

AUTHORS: Bayan Karimi et al

The additions and clarifications made by the authors addressed the points raised in the reviews satisfactorily. The manuscript in its revised form warrents publication.